# Using Artificial Neural Network Algorithm and Remote Sensing Vegetation Index Improves the Accuracy of the Penman-Monteith Equation to Estimate Cropland Evapotranspiration

**Yan Liu** [1] , **Sha Zhang** [1], **Jiahua Zhang** [2] , **Lili Tang** [1] **and Yun Bai** [1,*]

[1] Centre for Remote Sensing and Digital Earth, College of Computer Science and Technology, Qingdao University, Qingdao 266071, China; liuyan20210409@163.com (Y.L.); zhangsha@qdu.edu.cn (S.Z.); tanglili5208@163.com (L.T.)

[2] Aerospace Information Research Institute, Chinese Academy of Sciences, Beijing 100094, China; zhangjh@radi.ac.cn

\* Correspondence: baiyun@qdu.edu.cn; Tel.: +86-137-1880-5053

**Abstract:** Accurate estimation of evapotranspiration (ET) can provide useful information for water management and sustainable agricultural development. However, most of the existing studies used physical models, which are not accurate enough due to our limited ability to represent the ET process accurately or rarely focused on cropland. In this study, we trained two models of estimating croplands ET. The first is Medlyn-Penman-Monteith (Medlyn-PM) model. It uses artificial neural network (ANN)-derived gross primary production along with Medlyn's stomatal conductance to compute surface conductance ($G_s$), and the computed $G_s$ is used to estimate ET using the PM equation. The second model, termed ANN-PM, directly uses ANN to construct $G_s$ and simulate ET using the PM equation. The results showed that the two models can reasonably reproduce ET with ANN-PM showing a better performance, as indicated by the lower error and higher determination coefficients. The results also showed that the performances of ANN-PM without the facilitation of any remote sensing (RS) factors degraded significantly compared to the versions that used RS factors. We also evidenced that ANN-PM can reasonably characterize the time-series changes of ET at sites having a dry climate. The ANN-PM method can reasonably estimate the ET of croplands under different environmental conditions.

**Keywords:** evapotranspiration; penman-monteith equation; artificial neural network; canopy conductance

## 1. Introduction

Evapotranspiration (ET) is the process by which vegetation and groundwater transport water vapor to the atmosphere, mainly including plant transpiration and soil evaporation [1], with transpiration being dominant on a global scale [2]. Estimation of ET is an important basis for reasonable irrigation over croplands at a regional scale [3]; at the same time, as an important part of energy balance and the water cycle, ET also affects atmospheric circulation and plays an important role in regulating climate. Cropland is an important ecosystem on the land surface. Thus, the accurate estimation of cropland ET is of great significance for the rational irrigation of crops and the study of material and energy balance under the background of climate change [4].

The Penman–Monteith (PM) equation is the most commonly used framework for estimating regional or global ET. The regional-scale modeling process based on the PM equation is a simulation of surface conductance ($G_s$), and this parameter accounts for the largest source of uncertainty in ET modeling based on the PM equation on a regional scale. Cleugh et al. [5] tested two models of estimating land surface evaporation, the

surface energy balance model and PM-based approach using remote sensing (RS)-derived leaf area index (LAI), to estimate *Gs* at two Australian flux stations, and the PM-based method proved better. Mu et al. [6] found that the surface conductivity model of Cleugh et al. [5] was unreliable when used to estimate the global ET of 19 AmeriFlux sites due to the oversimplified estimates of surface conductance. Therefore, the canopy conductance and ET algorithms based on the PM method of Cleugh et al. [5] were improved by using the RS and global meteorological data. The algorithm of Mu et al. [6] considered the surface energy partitioning process and the environmental constraints of ET, but the performance of Mu et al. [6] still remains uncertain. Mu et al. [7] further improved the global terrestrial ET algorithm and showed the improved algorithm performed better compared to the original. Based on Cleugh et al. [5] and Mu et al. [6], Leuning et al. [8] developed a biophysical model to estimate *Gs* and introduced a simpler soil evaporation algorithm than the MOD16 algorithm [6] to calculate daily average evaporation. The results showed that the PM equation, incorporated with the RS leaf area index, could more reliably estimate the evaporation rate. However, the performances of the model degraded if a fixed value of maximum stomatal conductance ($g_{sx}$) was used to estimate the surface conductance across a wide range of vegetation categories [8]. Zhang et al. [9] further developed the *Gs* formula and calculated the land surface ET at a spatial resolution of 0.05 ° using the PM equation. Yebra et al. [10] reversed the PM equation to obtain the *Gs* of the plant canopy, and then the estimated *Gs* was used to retrieve actual ET using the parameterized PM equation. Kitao et al. [11] also applied a semi-empirical model dependent on photosynthesis [12] to estimate canopy *Gs*. Because the method of Ball et al. [12] restricted the applicability of the model, Yan et al. [13] used a simple biophysical model to calculate *Gs*, and then the computed *Gs* was used to calculated global ET based on the PM equation. Mallick et al. [14] estimated *Gs* by integrating the radiometric surface temperature into a combined structure of the PM model and the Shuttleworth–Wallace model and used the simplified surface energy balance model to estimate ET. The method of Yan et al. [13] used the leaf area index and surface meteorological data, while Mallick et al. [14] did not use any leaf-scale empirical parameter model to determine *Gs* and ET. However, the method of Mallick et al. [14] had a tendency to overestimate *Gs*. For areas with limited data, the method of Mallick et al. [14] was considered to be further improved. Therefore, Bhattarai et al. [15] used RS and reanalysis data to develop an automatic multi-model to estimate regional ET in important areas.

In order to reduce the uncertainties in ET estimation due to the difficulty in estimating *Gs*, semi-empirical models that use machine learning (ML) to more accurately calculate the *Gs* in the PM equation were proposed [16–18]. For example, Zhang et al. [18] combined ML, in which only temperature (Ta) data was used with the PM equation to estimate crop ET, and showed that the accuracy of the ML-based PM approach was better than the Hargreaves (HARG) method. However, the computational complexity of the model of Zhang et al. [18] is relatively high and requires more storage space. Traore et al. [17] evaluated different ML methods based on only temperature data to calculate ET under the framework of the PM equation. The determination coefficients ($R^2$) were significantly increased when wind speed data was added to the model of [17]. Thus, only one meteorological input is not enough for reasonably quantifying ET. Multiple data combinations can effectively improve the accuracy of the ET model. Zhao et al. [19] developed a hybrid model to estimate latent heat flux based on various variables (such as soil moisture, carbon dioxide concentration (Ca), etc.), combining ML models with the PM method. The results showed that the hybrid model is more adaptable to extreme environments compared with the pure ML method. Due to a lack of reliable and spatiotemporal continuous soil moisture data sets on a global scale, the model of Zhao et al. [19] is limited to a regional scale and cannot be applied on a global scale. Therefore, using only a single datum or a data set that is difficult to obtain will limit the application of the model on a regional or global scale. Therefore, we use a variety of globally available data combined with ML methods in order to improve the estimates of ET over croplands. The ML approaches can represent the complex and non-linearly

relationships between inputs and the target [20], and assess the adaptivity of multiple ET models of different environments [21], with smaller errors under a specific environmental condition.

Nowadays, most of the existing studies on estimating ET use physical models [22–25] or purely rely on ML algorithms [26–31]; these methods are not accurate enough to represent the ET due to the limited ability to understand the ET process. The hybrid ET model that combines the physical framework, namely the PM equation, and ML algorithms has proved to be effective in ET estimates [19,32]. The ML approaches resolved the difficulty of characterizing the complex environmental constraints on ET in the hybrid model, while the PM framework ensures the model's robustness. It is worth noting that the pure ML models may yield comparable or even better performance compared to the hybrid model [19] or individual physical models [26,33]. However, without physical constraints, the reliability of the pure ML models depends on the representativeness of training data [33]. As a result, the pure ML models are vulnerable to extreme environmental conditions [19], while the hybrid models show more robust performances under these conditions [19].

In this study, we aim to improve the estimates of cropland ET by training a hybrid ET model based on an artificial neural network (ANN) and PM equation, investigate whether the use of RS factors can improve the performances of hybrid models, and evaluate the ANN-PM model to simulate ET on a daily scale over flux sites covering a wide range of climate dryness.

## 2. Material and Methods

The research flow chart of this study is shown in Figure 1. We trained two methods to estimate ET. First, the *Gs* model is constructed using meteorological data and remote sensing data, and subsequently, used to simulate ET under the framework of the PM equation. Secondly, *Gs* is estimated using ANN-derived GPP in conjunction with Medlyn stomatal conductance, and then the computed *Gs* is used to estimate ET using the PM equation.

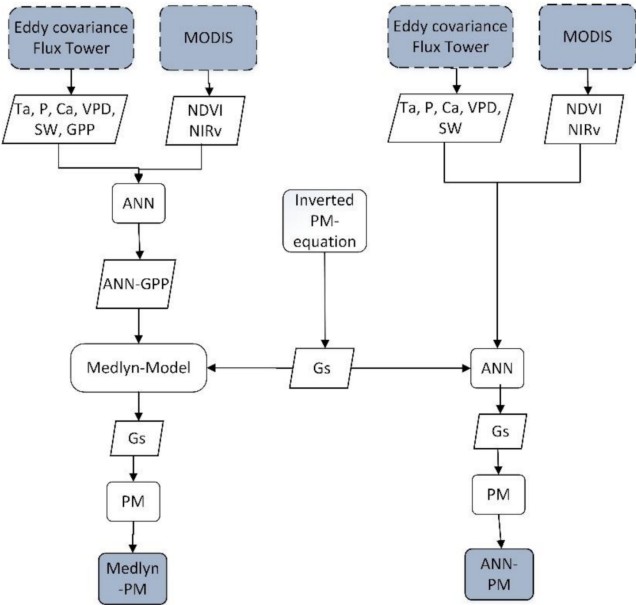

**Figure 1.** Research flow chart. Ta is temperature, P is precipitation, SW is solar radiation, Ca is carbon dioxide concentration, VPD is vapor pressure deficit, GPP is gross primary production, NDVI is normalized difference vegetation index, NIRv is near-infrared reflectance of vegetation, ANN is artificial neural network, *Gs* is surface conductance, and PM is the Penman–Monteith equation. A white parallelogram denotes a variable, and a white rectangle denotes a method. A gray dotted rectangle denotes the source of the variable, and a gray solid rectangle denotes a model.

### 2.1. Material

The meteorological data used in this study were retrieved from the meteorological observation data of the eddy covariance flux tower at 17 flux sites. Figure 2 shows the map representation of the 17 flux sites of cropland over the globe.

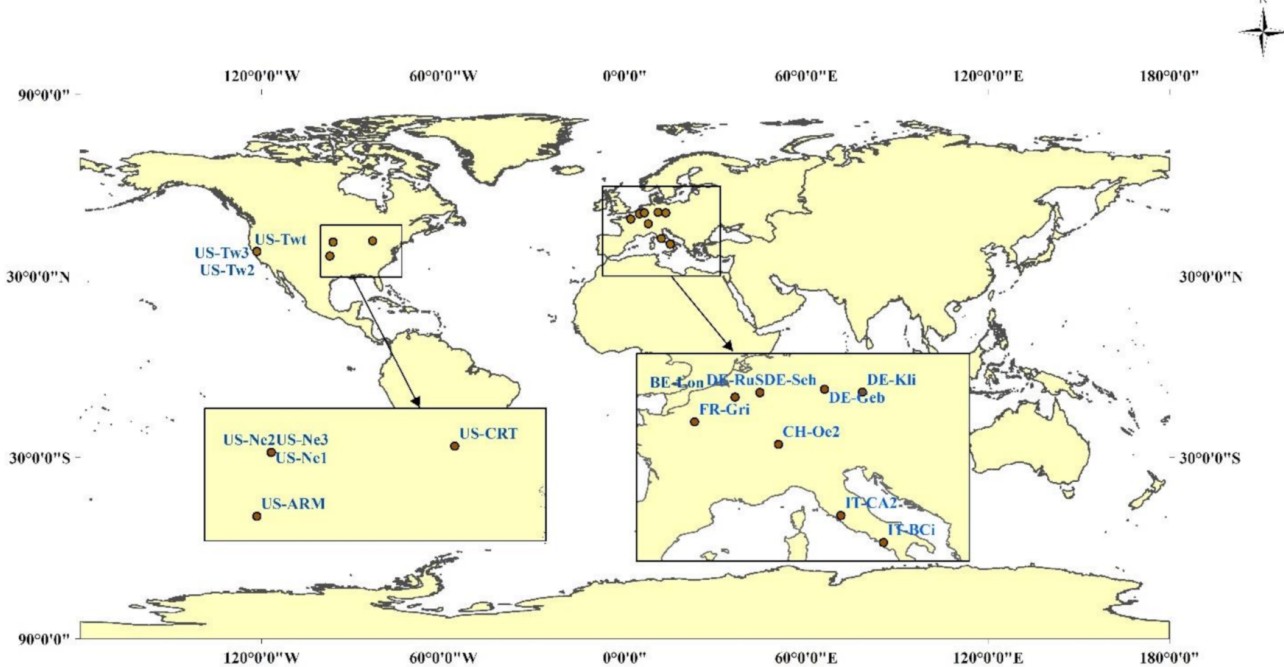

**Figure 2.** Map representation of 17 eddy covariance flux sites.

The information of the 17 flux sites is shown in Table 1. The 17 flux sites of cropland over the globe were located in different countries (such as Germany, the United States, France, and Italy). DE-Kli and IT-BCi have the lowest (7.77 °C) and highest (17.88 °C) mean annual temperatures, respectively. The annual precipitation of these sites varies from 343.1 (US-Tw3) to 2062.25 mm (CH-Oe2). We divide the flux data set into the training set, validation set, and test set, the ratios of which are 60%, 20%, and 20%, respectively, and the three datasets are used to train, validate, and test the ANN model. The vegetation index and reflectance data were retrieved from MODIS MOD09A1 (https://modis.ornl.gov/data.html, accessed on 27 February 2020), having a spatial resolution of 500 m. These flux data and MODIS data were used to training the two models of estimating ET. The time series of MODIS data were extracted according to the longitude and latitude coordinates of the flux sites. The spectral index was calculated using the MOD43A4 product, following the formulations shown in Table 2. NDVI is usually used to reflect the information of vegetation coverage and growth. In order to obtain information on a larger regional scale, a new vegetation index NIRv is introduced [19], which can reflect the photosynthetic capacity of surface vegetation better. NIRv is the product of the total near-infrared reflectance (NIRt) (MODIS second band) and NDVI. NIRv is a remote sensing measurement of canopy structure, which can more accurately predict photosynthesis [34]. The shortwave infrared band (SWIR) is usually used to reflect water stress and is calculated by using the reflectance data directly.

**Table 1.** Description of flux sites.

| Site Code | Latitude | Longitude | Mean Annual Temperature (°C) | Mean Annual Precipitation (mm) | Years | Reference |
|---|---|---|---|---|---|---|
| BE-Lon | 50.55 | 4.75 | 11.41 | 766.50 | 2004–2014 | Moureaux et al. [35] |
| CH-Oe2 | 47.29 | 7.73 | 9.56 | 2062.25 | 2004–2014 | Moors et al. [36] |
| DE-Geb | 51.10 | 10.91 | 9.67 | 532.90 | 2001–2014 | Anthoni et al. [37] |
| DE-Kli | 50.89 | 13.52 | 7.77 | 810.30 | 2004–2014 | Brust et al. [38] |
| DE-RuS | 50.86 | 6.45 | 10.80 | 551.15 | 2011–2014 | Eder et al. [39] |
| DE-Seh | 50.87 | 6.45 | 10.29 | 573.05 | 2007–2010 | Korres et al. [40] |
| FR-Gri | 48.84 | 1.95 | 10.96 | 598.60 | 2004–2014 | Loubet et al. [41] |
| IT-BCi | 40.52 | 14.96 | 17.88 | 1197.20 | 2004–2014 | Ranucci et al. [42] |
| IT-CA2 | 42.38 | 12.03 | 14.84 | 766.50 | 2011–2014 | |
| US-ARM | 36.60 | −97.49 | 15.27 | 646.05 | 2003–2012 | Raz-Yaseef et al. [43] |
| US-CRT | 41.63 | −83.35 | 10.85 | 810.30 | 2011–2013 | Chu et al. [44] |
| US-Ne1 | 41.1651 | −96.477 | 10.54 | 846.80 | 2001–2013 | Verma et al. [45] |
| US-Ne2 | 41.1649 | −96.470 | 10.26 | 876.00 | 2001–2013 | Suyker and Verma [46] |
| US-Ne3 | 41.1797 | −96.440 | 10.38 | 697.15 | 2001–2013 | Suyker and Verma [46] |
| US-Tw2 | 38.1047 | −121.643 | 15.23 | 386.90 | 2012–2013 | Knox et al. [47] |
| US-Tw3 | 38.1159 | −121.647 | 16.00 | 343.10 | 2013–2014 | Baldocchi et al. [48] |
| US-Twt | 38.1087 | −121.653 | 14.75 | 357.70 | 2009–2014 | Hatala et al. [49] |

**Table 2.** Calculation of vegetation index. *rx* represents the reflectivity of MODIS bands (*x* = 1–7), NDVI is the normalized difference vegetation index, NIRv is near-infrared reflectance of vegetation.

| Index | Formula |
|---|---|
| NDVI | $NDVI = \frac{r2 - r1}{r2 + r1}$ |
| NIRv | $NIRv = NDVI * r2$ |

### 2.2. Two ET Models Based on ANN

In this study, two models were trained based on the PM equation, and the difference lies in the *Gs* calculation. The following two summaries introduce the two methods in detail. The formula of the PM equation is as follows:

$$\lambda E = \frac{(Rn - G) \cdot \Delta + \rho \cdot Cp \cdot D \cdot Ga}{\Delta + \gamma(1 + Ga/Gs)} \tag{1}$$

where $\lambda E$ is evapotranspiration, *Rn* is net radiation, *G* is soil heat flux, $\Delta$ is the gradient of the saturation vapor pressure versus atmospheric temperature, $\rho$ is air density, *Cp* is the specific heat at constant pressure of air, *D* is the vapor pressure deficit of the air, *Ga* is the aerodynamic conductance, and $\gamma$ is the psychometric constant.

In order to test the effects on the accuracy of using different combinations of input variables, different combinations of input variables in the ANN are shown in Table 3.

**Table 3.** Different combinations of input variables in the ANN. Ta is temperature, P is precipitation, SW is solar radiation, Ca is carbon dioxide concentration, VPD is vapor pressure deficit, NDVI is normalized difference vegetation index, NIRv is near-infrared reflectance of vegetation, and SWIR is shortwave infrared band.

| Number | Input Parameters |
|---|---|
| 1 | Ta, P, SW, Ca, VPD |
| 2 | Ta, P, SW, Ca, VPD, NDVI |
| 3 | Ta, P, SW, Ca, VPD, NDVI, NIRv |
| 4 | Ta, P, SW, Ca, VPD, NDVI, NIRv, SWIR |

### 2.2.1. ANN-PM Model

We trained an ANN-PM model based on ANN and PM equations to estimate ET. ANN is a commonly used ML method, which has been widely used in estimating ET. It consists of a large number of nodes, called neurons, which are connected to each other. The typical structure of ANN used to estimate ET is shown in Figure 3.

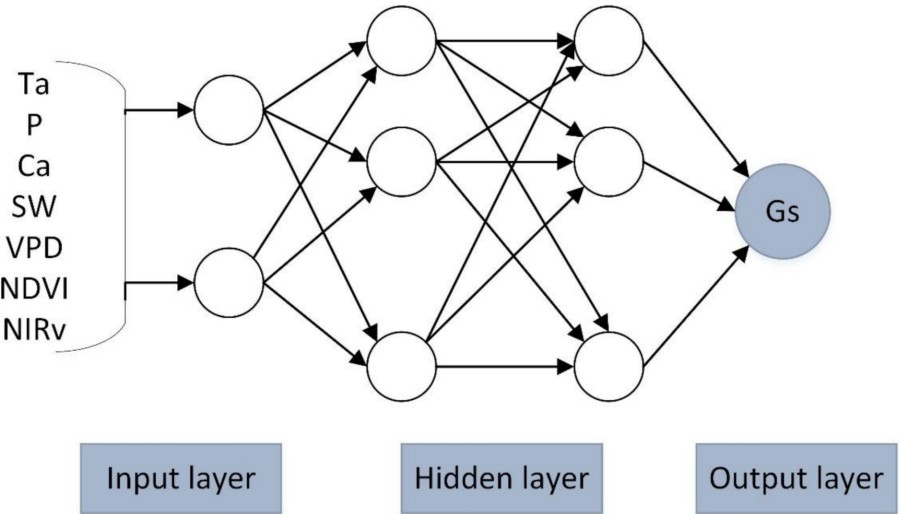

**Figure 3.** The typical structure of ANN.

ANN contains three layers: the input layer, hidden layer, and output layer. The input layer is responsible for receiving input data, the hidden layer constructs the relationships between the input and output, and the output layer outputs the predicted target values. The variables input to ANN in this study includes Ta, precipitation (P), solar radiation (SW), Ca, vapor pressure deficit (VPD), normalized difference vegetation index (NDVI), and near-infrared reflectance of vegetation (NIRv). In the variables we used, Ta, SW, Ca, and VPD can affect canopy conductivity from different aspects [50]. The consideration of P is mainly to represent the influence of canopy interception on ET. Thus they are selected to model *Gs*. There is an interaction and mutual influence between the transpiration and photosynthetic capacity of plants, and ET is dominated by transpiration. The vegetation index, NIRv, is considered in order to better reflect the impact of the photosynthetic capacity of the surface vegetation on evapotranspiration. NIRv is able to characterize seasonable variations in canopy scale photosynthesis rate without additional environmental factors that are conventionally used to constrain photosynthesis [34]. These variables are used to train ANN to the *Gs* model. Referring to Zhao et al. [19], we used the ANN model to model ln(*Gs*) rather than *Gs* because the logarithmic form can effectively reduce the effect of errors in *Gs* calculated from the observations. Finally, the logarithm of *Gs* obtained by ANN simulation is converted to *Gs*, and then the converted *Gs* is input into the PM equation to calculate ET. Here, *Gs* values used to train the ANN model were calculated from the observed ET along with the inverted PM equation [51]. In order to avoid over-fitting, the network model is repeatedly trained, where the number of hidden layers ranges from 1 to 10, and the number of neurons in each layer increases from 1 to 128, with an interval of 8. Then, we choose the optimal ANN structure as the best model.

### 2.2.2. Medlyn-PM Model

The Medlyn-PM model uses ANN-derived GPP in conjunction with a theoretical *Gs* model to estimate surface conductance, and then the computed *Gs* is used to estimate ET using the PM equation. Firstly, we use the optimal ANN structure selected above to train the GPP model. Secondly, on the pixel scale, the computed GPP, Ca, and air vapor pressure deficit are used for *Gs* regression analysis to establish the relationship among them and determine the undetermined coefficients $g_0$ and $g_1$. Then, we use the above variables and

the relationship between them to build the *Gs* model. Finally, the constructed *Gs* is input into the PM equation to calculate ET. The relationship is as follows [52]:

$$Gs = 1.6 * \frac{GPP}{Ca} * \left( \frac{g_1}{\sqrt{D}} + 1 \right) + g_0 \tag{2}$$

where *Gs* is stomatal conductance, GPP is gross primary production, Ca is $CO_2$ concentration of the air, $g_1$ and $g_0$ are undetermined coefficients derived from regression analysis, and D is the vapor pressure deficit of the air. The minimum value of D is fixed to 0.1 KPa.

### 2.3. ANN Architecture Optimization

The ML method, i.e., ANN, used in the ANN-PM and the Medlyn-PM, considers input variables, including Ta, P, SW, Ca, VPD, NIRv, and NDVI. Usually, in order to reduce over-fitting, the network model is repeatedly trained. Thus, we need to recognize the best ANN structure. In our study, the optimal ANN is determined in terms of mean square error (MSE) while minimizing the number of degrees of freedom based on the Akaike Information Criterion (AIC). AIC is a standard to measure the goodness of fit of the statistical model. AIC encourages the goodness of data fitting but tries to avoid over-fitting. Therefore, the priority model should be the one with the lowest AIC value. Cropland ET is estimated by combining the predictive output of ANN with the PM equation. The calculation formula of the AIC indicator is as follows [53]:

$$AIC = \log(MSE) + \frac{2q}{n} \tag{3}$$

where MSE is mean square error, *q* is the total number of parameters in the network, and *n* is the number of observations in the training sample.

### 2.4. Model Evaluation

#### 2.4.1. Model Performance Measurement

The model performance evaluation metrics used in the study include root mean square error (RMSE), mean absolute error (MAE), and determination coefficients ($R^2$). The calculations of these metrics are shown in Table 4.

**Table 4.** Calculation formula of evaluation parameters. RMSE is the root mean square error, MAE is the mean absolute error, and $R^2$ is the determination coefficients. *fi*: Predicted value: $\overline{fi}$ Mean value of the predicted values; *yi*: Experiment value; $\overline{yi}$: Mean value of the observed values; *m*: Total amount of experimental data.

| Evaluation Parameters | Formula |
| :---: | :---: |
| RMSE | $\sqrt{\frac{\sum_{i=1}^{m}(fi-yi)^2}{m}}$ |
| MAE | $\frac{\sum_{i=1}^{m}|fi-yi|}{m}$ |
| $R^2$ | $\left( \frac{\sum_{i=1}^{m}(yi-\overline{yi})(fi-\overline{fi})}{\sqrt{\sum_{i=1}^{m}(yi-\overline{yi})^2}\sqrt{\sum_{i=1}^{m}(fi-\overline{fi})^2}} \right)^2$ |

RMSE is the standard deviation between the predicted and true values, reflecting the degree that the predicted values explain the true values [54]. MAE is the mean error of evaluating a set of predictions and is the average value of the absolute difference between predicted and experimental values on test samples, but MAE is less sensitive to extreme values than RMSE [55]. $R^2$ is determined by drawing a scatter plot between the observed and predicted value. Lower RMSE, MAE, and higher $R^2$ correspond to a better performance of the model.

2.4.2. Evaluating the Model Used to Estimate ET under Dry Climate

Modeling ET in dry regions is more challenging than in other regions, especially for croplands. Because the water status of croplands is affected by irrigation, and the information of irrigation on a regional scale is difficult to obtain. On the other hand, in arid areas, most of the precipitation is consumed in the process of ET, and inadequate water supply could substantially limit the growth of crops in these regions. Therefore, accurate estimation of ET plays an important role in the sustainable development of agriculture in arid areas. Research on modeling ET in dry climates can facilitate rational cropland irrigation, maintaining stable crop production in dry regions.

We analyzed the performance of the models we trained in estimating ET under a dry climate. The aridity index (AI) is a means and tool to determine the drought degree and range of a certain period quantitatively, and it is also an indicator of the degree of dry and wet in a region. The calculation formula of the AI is as follows [56]:

$$AI = \frac{P}{PET} \tag{4}$$

where AI is aridity index, PET is potential evapotranspiration, and P is the average precipitation. The AI calculation of each site is limited to the time range covered by the site. Low AI corresponds to a dry climate. We selected the sites with the AI values below 0.5 as arid areas by calculating the AI values of each flux site.

## 3. Results

### 3.1. Model Parameter Optimization

The undetermined parameters $g_0$ and $g_1$ were required for running Medlyn-PM. They were determined by fitting the analytical *Gs* equation, $Gs = 1.6 * \frac{GPP}{Ca} * \left( \frac{g_1}{\sqrt{D}} + 1 \right) + g_0$, and we obtained that $g_0 = 0.06$ and $g_1 = 3.94$. The variations in RMSE/MAE/$R^2$ with the change of the numbers of hidden layers and neurons for the ANN-PM model with training and validation datasets are presented in Figure 4.

The figure shows that the RMSE and MAE of ANN-PM with the training dataset decrease gradually as the number of hidden layers (HL) and the number of neurons increase. The RMSE and MAE of ANN-PM with the validation dataset decrease as the numbers of hidden layers (HL) and neurons increase from 1 (the number of HL) −1 (the number of neurons) to 10–48 but increase after the number of the two parameters become larger than 1–48. As the number of hidden layers and the number of neurons increase to 10–128, the $R^2$ of the training dataset reaches a maximum value (0.94), and the $R^2$ of the validation dataset is concentrated around 0.80. Then, considering the AIC values, we identified the best architectures of ANN-PM (AIC = −0.76) and Medlyn-PM (AIC = −0.55) models and the key parameters are shown in Table 5. The ANN-PM model has an ANN structure with two hidden layers and 48 neurons in each layer. The AIC index is also used to select the ANN-based GPP model in Medlyn-PM, and the optimal model has two hidden layers and one neuron in each layer.

**Table 5.** The key parameters of the two models. ANN is artificial neural network and PM is the Penman-Monteith.

|  | ANN-PM Model | Medlyn-PM Model |
|---|---|---|
| The number of hidden layers | 2 | 2 |
| The number of neurons | 48 | 1 |

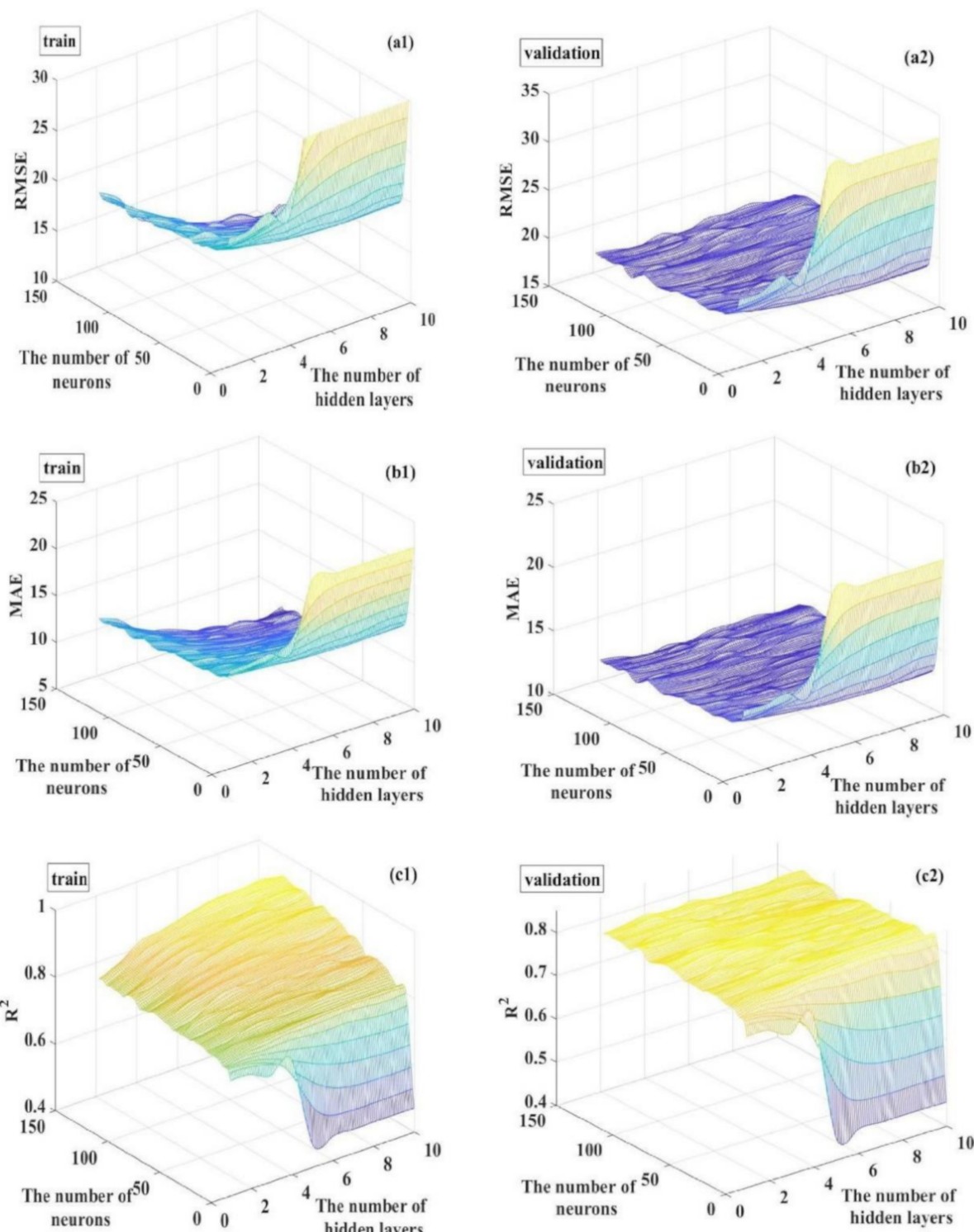

**Figure 4.** A three-dimensional graph between the number of hidden layers, the number of neurons, and RMSE/MAE/$R^2$ of the training and validation datasets of the ANN-PM model. (**a1**) is the RMSE of the training, (**a2**) is the RMSE of the validation, (**b1**) is the MAE of the training, (**b2**) is the MAE of the validation, (**c1**) is the $R^2$ of the training, (**c2**) is the $R^2$ of the validation. RMSE is the root mean square error, MAE is the mean absolute error, and $R^2$ is the determination coefficient.

### 3.2. Comparison of ANN Model with Different Input Data

The input data of ANN in the ANN-PM model includes meteorological data (Ta, P, SW, Ca, and VPD) and remote sensing data (NIRv and NDVI). We investigate the accuracy

of estimating ET using the optimized ANN-PM (two hidden layers and 48 neurons in each layer) with several combinations of input data (Table 3). Figure 5 shows the comparisons between the predicted ET values and the measured values of cropland ET in the training, validation, and test datasets across all flux sites.

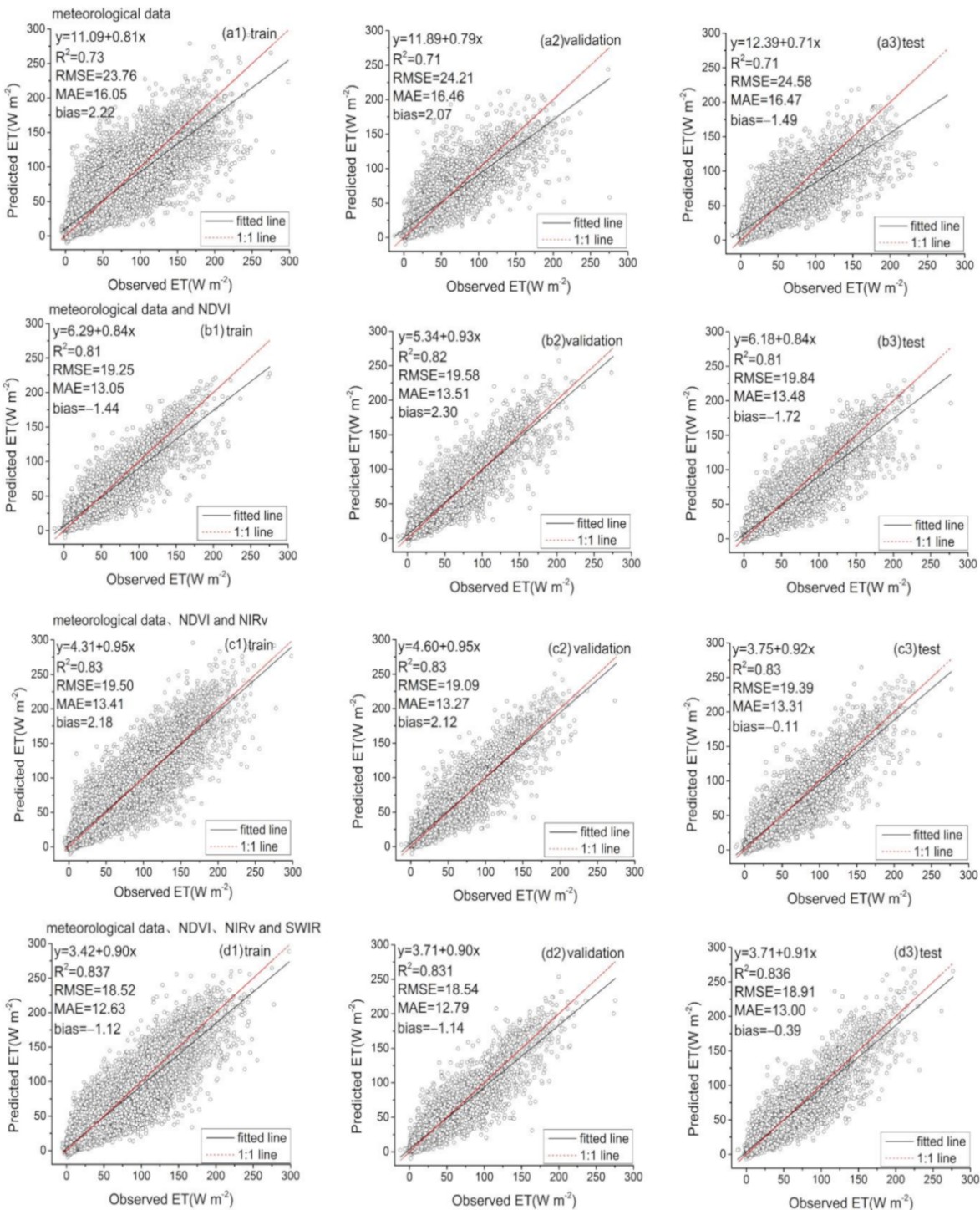

**Figure 5.** Scatter plots between the predicted ET values and the observed ET values measured from the flux tower in the training, validation, and test datasets of the ANN-PM model. (**a1**–**a3**) is the scatter plot between the predicted ET values and the observed ET values measured from the flux tower of the ANN-PM model using meteorological data in the three datasets, (**b1**–**b3**) is the scatter plot using meteorological data and NDVI, (**c1**–**c3**) is the scatter plot using meteorological data and NDVI and NIRv, (**d1**–**d3**) is the scatter plot using meteorological data and NDVI and NIRv and SWIR.

As shown in Figure 5, all the employed models provide different accuracies under different input combinations. The accuracy of predicted ET values differs significantly depending on the model types and input combinations. Except for the second input combination, all input combinations show the highest $R^2$ in the training stage (Figure 5(a1,c1,d1)). The ranks of the input combinations under investigation in terms of prediction accuracy are (the value in parentheses after RMSE indicates the percentage of RMSE relative to the observed value): the fourth input combination ($R^2$ = 0.831–0.837, RMSE = 18.52–18.91 W m$^{-2}$ (38.42–38.86%), MAE = 12.63–13.00 W m$^{-2}$), the third input combination ($R^2$ = 0.83, RMSE = 19.09–19.50 W m$^{-2}$ (39.84–40.46%), MAE = 13.27–13.41 W m$^{-2}$), the second input combination ($R^2$ = 0.81–0.82, RMSE = 19.25–19.84 W m$^{-2}$ (39.94–41.05%), MAE = 13.05–13.51 W m$^{-2}$), and the first input combination ($R^2$ = 0.71–0.73, RMSE = 23.76–24.58 W m$^{-2}$ (49.29–50.75%), MAE = 16.05–16.47 W m$^{-2}$). In the testing stage, the models of the third input combination and fourth input combination have identical performance in estimating ET, both of which performed superior to the second input combination and the first input combination in predicting ET. These results confirm that the model using all input variables (meteorological data and three remote sensing data factors (NDVI, NIRv, SWIR)) achieves the best performances (RMSE = 18.52–18.91 W m$^{-2}$ (38.42–38.86%), MAE = 12.63–13.00 W m$^{-2}$, and $R^2$ = 0.831–0.837) compared with those using a subset of all the variables. However, the model using meteorological data and two remote sensing data factors (NDVI and NIRv) is also capable of predicting ET with acceptable accuracy, having the RMSE and MAE values of 19.09–19.50 W m$^{-2}$ (39.84–40.46%) and 13.27–13.41 W m$^{-2}$, respectively. When using only meteorological data, the model shows degraded performance with larger errors (RMSE = 23.76–24.58 W m$^{-2}$ (49.29–50.75%) and MAE = 16.05–16.47 W m$^{-2}$) and smaller determination coefficients ($R^2$ = 0.71–0.73). The model using the combination of meteorological data and one remote sensing factor, NDVI, shows intermediate results (RMSE = 19.25–19.84 W m$^{-2}$ (39.94–41.05%), MAE = 13.05–13.51 W m$^{-2}$, and $R^2$ = 0.81–0.82). The model using meteorological data and three remote sensing data factors (NDVI, NIRv, and SWIR) showed comparable performance with that using meteorological data and two remote sensing data factors (NDVI, NIRv). Therefore, it can be concluded that remote sensing data in the ANN model facilitated the improvement of the estimates of croplands ET.

### 3.3. Comparison of ANN-PM and Medlyn-PM

Figure 6 shows the scatter plots of measured ET vs. predicted ET by the Medlyn-PM and the ANN-PM model, respectively. At the site scale, the two models differ substantially in performance from each other. Figure 6 shows good correlations between the observed ET and the predicted ET by the two methods ($R^2$ = 0.75 and 0.83). Figure 6 also illustrates that the $R^2$ value of the ANN-PM model is 0.08–0.09 higher than that of the Medlyn-PM model and the RMSE and MAE of ANN-PM are 4.26–4.3 and 3.12–3.34 W m$^{-2}$ smaller than that of the Medlyn-PM model, respectively. Overall, the ANN-PM model shows relatively high accuracy with smaller RMSE and MAE, and larger $R^2$ (RMSE = 19.09-19.50 W m$^{-2}$ (39.84–40.46%), MAE = 13.27–13.41 W m$^{-2}$, $R^2$ = 0.83) in estimating cropland ET compared to the Medlyn-PM model (RMSE = 23.39–23.76 W m$^{-2}$ (49.95–51.14%), MAE = 16.39–16.75 W m$^{-2}$, and $R^2$ = 0.74–0.75), indicating a great advantage in estimating cropland ET using the ANN-PM model.

### 3.4. Accuracy of ANN-PM Model under Dry Climates

In arid areas, most of the precipitation is consumed in the process of ET, and inadequate water supply could substantially limit the growth of crops in these regions. Therefore, accurate estimation of ET plays an important role in the sustainable development of agriculture in arid areas. Hence, we evaluated the ANN-PM model to simulate ET on a daily scale over flux sites covering a wide range of climate dryness, measured using aridity index (AI). The $R^2$ between simulation and observation is used to measure the model performance. The variations in $R^2$ of each flux site in relation to site-scale AI are shown in Figure 7, where

low AI values correspond to dry climates. The driest site is US-Twt, followed by US-Tw3, US-Tw2, and DE-Rus. The average $R^2$ of the 16 flux sites is 0.74, and the average $R^2$ of the driest four flux sites with an AI index lower than 0.5 (DE-Rus = 0.49, US-Tw2 = 0.42, US-Tw3 = 0.30, and US-Twt = 0.26) is 0.77. In terms of $R^2$, the performances of the ANN-PM model at the dry sites are reasonable and comparable to those at the wet sites (Figure 7).

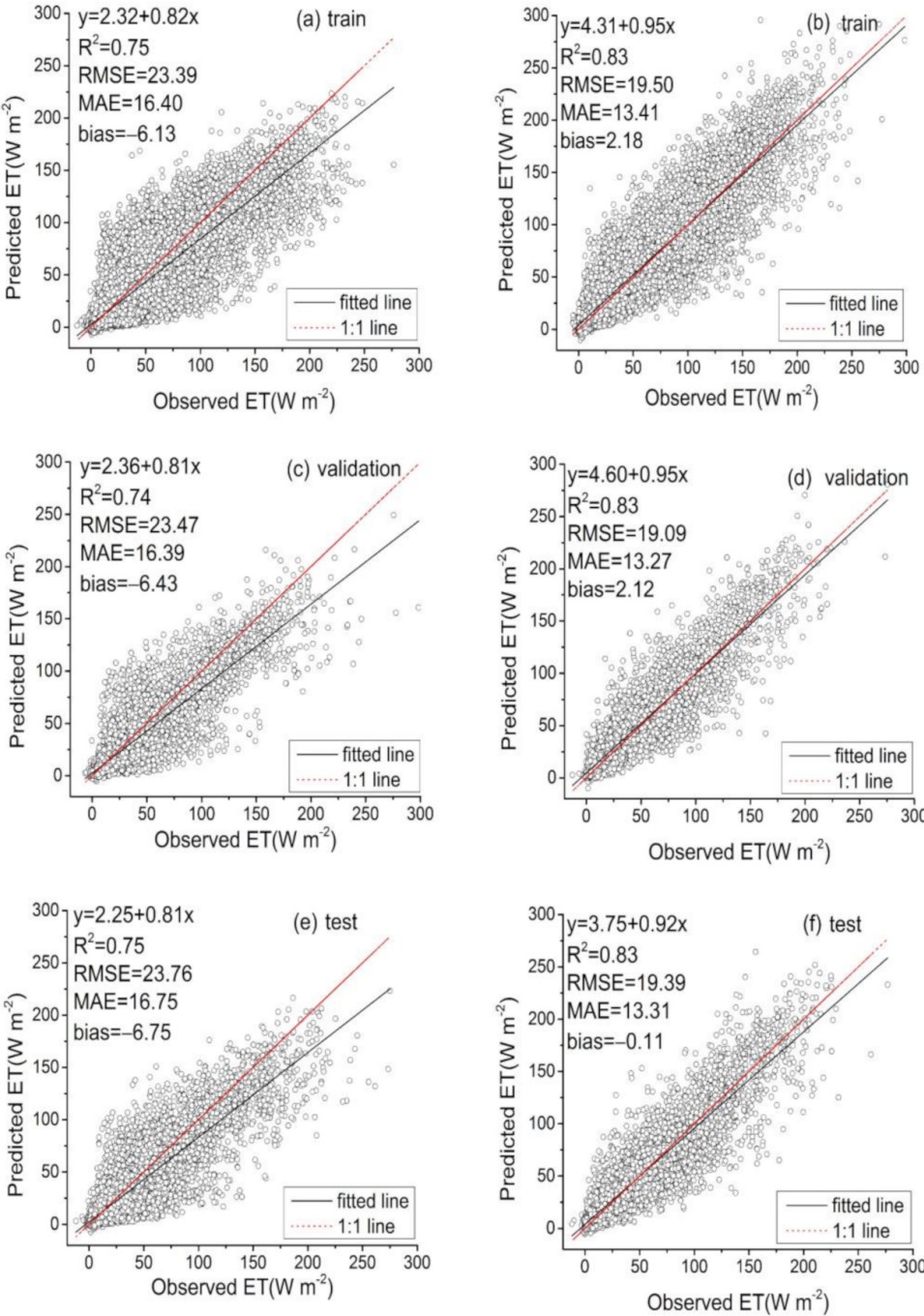

**Figure 6.** Scatter plots of the observed ET values measured from the flux tower and predicted ET values of the Medlyn-PM (**left**) and the ANN-PM model (**right**) in estimating cropland evapotranspiration. (**a,c,e**) are the scatter plots of the observed ET values measured from the flux tower and predicted ET values of the Medlyn-PM model in the training, validation, and test datasets, respectively. (**b,d,f**) are the scatter plots of the observed ET values measured from the flux tower and predicted ET values of the ANN–PM model in the training, validation, and test datasets, respectively.

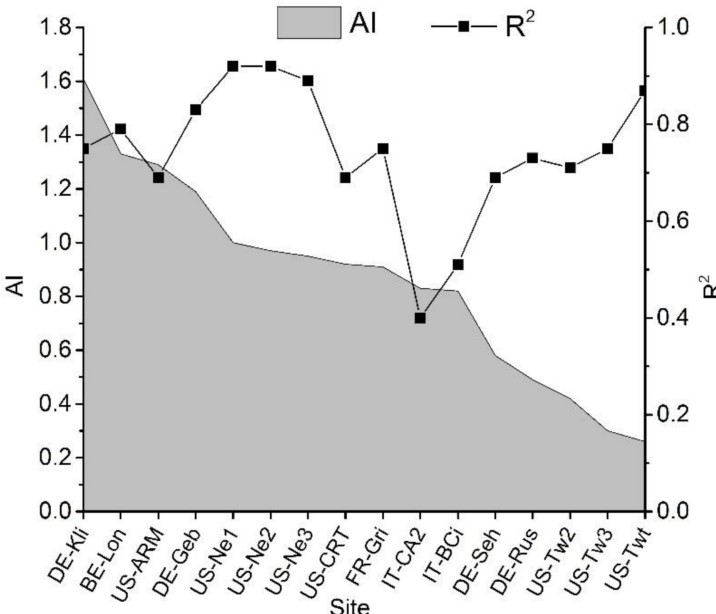

**Figure 7.** AI and $R^2$ values of each flux site. AI is aridity index and $R^2$ is the determination coefficients between simulation and observation.

The ANN-PM model can capture the time-series changes of ET at the dry sites well (Figure 8, four sites with an AI index lower than 0.5). At the driest site, US-Twt, which is a paddy field site, ET predicted by the ANN-PM model agreed well with the observations, indicating that the model can reflect the influence of irrigation on cropland ET under dry conditions. Consequently, the ANN-PM model can simulate cropland ET across a wide range of gradients of climate dryness, showing great potential to estimate cropland ET accurately on a regional scale.

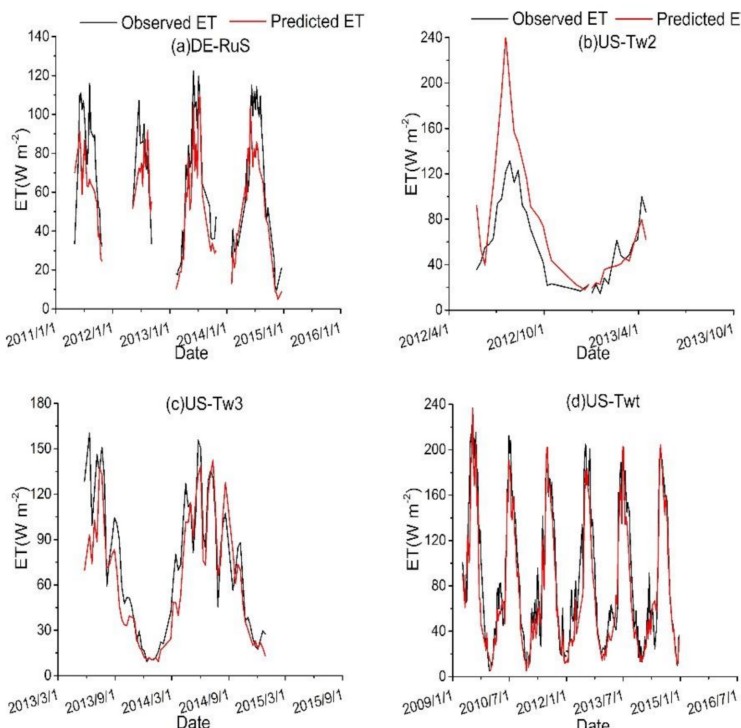

**Figure 8.** Time-series diagrams of observed ET (**black line**) measured from the flux tower and simulated ET (**red line**) by the ANN-PM model.

## 4. Discussion

### 4.1. Discussion of the Number of Sites

We used 17 sites in our study, and the time span of all sites is 2001–2014 (Table 1). The entire dataset contains more than 50,000 samples on a daily scale, which are large enough for establishing the ML-based method. As we know, the size of the sample we used is larger than some existing publications. For example, Zhu et al. [31] used nine stations in the arid region of Northwest China during the period 2002–2016. Yin et al. [57] evaluated ET in the eddy covariance flux observations at 14 Chinese flux tower sites during the period 2003–2017, and each site has at least 3 years of reliable data. Hossein Kazemi et al. [58] only used the daily meteorological records of seven weather stations in Iran for 10 years (2008–2017). Therefore, our data are enough to train a machine learning model. Our study is mainly for cropland. There are currently limited open-access cropland sites, but our sites cover the current main farming areas. These areas cover different climate types. Therefore, our model has wide applicability. There is currently a lack of stations in tropical regions. When applied in this climate region, the model needs to be further tested.

### 4.2. Comparison between This Research and Existing Research

The ANN-PM model of this study combines ML methods and the PM equation, and the remote sensing data of inputting into ANN contains a recently proposed NIRv index, which can be used to reflect the photosynthetic capacity and water status of the surface vegetation. Combining NIRv with ML and the PM equation shows great advantages in estimating cropland ET. Zhao et al. [19] used an ML method (ANN) and PM equation to estimate ET, but the study used soil moisture data that is difficult to obtain, which limits the application of the model in a large-scale and long-term series. Yamaç and Todorovic [59] combined the PM equation with three ML methods (K nearest neighbor algorithm, ANN, and Adaptive Boosting model) to estimate the ET using available weather input data with four different scenarios (temperature, solar radiation, wind speed, and relative humidity). They showed that using the combination of four data scenarios performs better than any other combinations. The above two studies are based on the theoretical framework of the PM equation and use ML methods. However, the first study uses soil moisture data that is not feasibly accessed on a regional scale, and the second uses only meteorological data, which is only applicable in a limited area. Compared with the above two studies, we combined meteorological data with remote sensing data to estimate ET. The fitting effect is better, and accuracy is improved. The model tested was applicable to a wide range of environmental gradients. He et al. [60] used a process and PM-based ET model, the MOD16 algorithm, to estimate ET for cropland sites (US-Tw2, US-Tw3, and US-Twt). The results showed that the site US-Tw2 has a higher $R^2$ (0.72) than US-Tw3 and US-Twt. In our study, we evaluated the performance of our ET models at three cropland sites (US-Tw2, US-Tw3, and US-Twt), respectively. Compared with He et al. 's [60] study, our models at the three sites all show higher accuracy ($R^2$ = 0.74–0.86). Our hybrid model, based on ML and PM, can perform better than the model based on the process and PM equation. Amazirh et al. [61] used the PM equation to estimate ET in semi-arid areas by introducing a simple relationship between surface resistances ($r_c$) and verified the model at flood and drip irrigation sites. The results showed that the $R^2$ of these two sites were 0.76 and 0.70, respectively, and the RMSEs were 22 and 23 W m$^{-2}$, respectively. Feng et al. [62] compared the performance of the PM equation and self-optimizing nearest neighbor algorithm (CCA-k-NN) in estimating ET. The results showed that the performance of CCA-k-NN was comparable with PM ($R^2$ = 0.8, RMSE = 24.01 W m$^{-2}$, MAE = 18.06 W m$^{-2}$). The above studies only used the PM equation to estimate cropland ET. Our study combines ML methods with the PM equation to estimate cropland ET ($R^2$ = 0.84, RMSE = 17.40 W m$^{-2}$, MAE = 12.41 W m$^{-2}$), the estimating accuracy obtained in this study is better, and the physical mechanism of the PM equation can ensure that the simulation result is always within the range of potential evapotranspiration.

### 4.3. Comparison of the ANN-Based ET Model with Existing ML-Based ET Models

ML algorithms have been more and more widely used to estimate ET on a regional or global scale. In this study, the most widely used ANN algorithm is used to improve the accuracy of the PM equation to estimate cropland ET on a regional scale. There are also many studies that use other ML algorithms to estimate ET, e.g., Abdullah et al. [63], Antonopoulos and Antonopoulos [64], Reis et al. [29], Yamaç and Todorovic [59], Zhu et al. [31], and Ferreira and da Cunha [65]. These studies literally showed different performances of different ML-based ET models. However, it should also be noted that the performance metrics of ET models could vary between different regions, validation data sources, temporal scale of validation, and so on. For example, the ML models estimating the reference ET usually show higher performance metrics than the actual ET models [64,66,67], as reference ET was calculated from only a few meteorological factors. If different data sources are used in modeling ET using the ML algorithm, the efficiency of the ET model can also be different. For example, Fan et al. [67] showed that the performance of the ML algorithm ($R^2$ = 0.701–0.995, RMSE = 0.106–0.637 mm d$^{-1}$) in estimating reference ET were significantly different between eight meteorological stations that represented the eight main climate types of China. Zhu et al. [31] showed similar results in modeling reference ET using the ML over nine meteorological stations in the arid region of Northwest China ($R^2$ = 0.844–0.969, MAE = 0.268–0.635 mm d$^{-1}$). The ET model focusing on the daily scale also produces different performance metrics from the hourly scale ET model. Ferreira and da Cunha [65] revealed better performances of the deep learning-based models in estimating daily reference ET on a daily scale as compared to the models on an hourly scale, with $R^2$ increased from 0.78–0.88 to 0.87–0.91, and RMSE decreased from 0.56–0.73 to 0.47–0.60 mm d$^{-1}$. The above studies show that the performance of the ET models can differ under different temporal scales. The performance metrics of the hybrid model in our study are in line with the range of those ML-based ET models.

### 4.4. The Reasons for the Low Accuracy of the Medlyn-PM Model and the Lack of the ANN-PM Model

The reason for the degraded performance of Medlyn-PM in estimating cropland ET, as compared to ANN-PM, is that the effect of soil evaporation is not considered in the model. ET includes soil evaporation and plant transpiration, as well as part of the contribution of canopy interception. Soil evaporation cannot be ignored in ET. Yu et al. [68] investigated the contribution of soil evaporation to ET of winter wheat under sprinkler irrigation. Their results showed that soil evaporation was an important part of ET, accounting for 20–28% of ET. Liu et al. [69] used a large-scale weighing permeameter and a micro permeameter to measure the daily evaporation and ET in winter wheat fields, and the study showed that soil evaporation accounted for 30% of the ET. Qin et al. [70] also showed that evaporation accounted for 32% of the total ET during the growth of winter wheat and 65% in the early growth period. These indicated a considerable contribution of soil evaporation in ET. Since ANN-PM used ANN to estimate the bulk surface conductance, which accounts for the effect of both stomatal and soil conductance, it has been found to perform better than the Medlyn-PM model.

The remote sensing information allows ANN-PM to simulate spatiotemporally continuous ET information [71]. However, we did not exhaust all possible RS data in the ANN-PM, which is beyond the scope of this study. In the future, we can evaluate more RS data to improve the accuracy of the ANN-PM model. For example, the development of multi-source RS data and surface parameter inversion products can provide PM models with some basic parameters that promote their application [72], so multi-source remote sensing data and PM models can be combined to estimate cropland ET.

## 5. Conclusions

The accurate estimation of cropland ET is important for crop irrigation, fertilization, and other management measures. In this study, we proposed an ANN-PM model based on ML and the PM equation to estimate cropland ET. At the same time, we optimized the Medlyn-PM model (uses ANN-derived GPP along with Medlyn's stomatal conductance to compute $Gs$, and the computed $Gs$ is used to estimate ET). We compared the two models to get a better method for estimating ET based on the ML approach. Specifically, we used ANN to estimate $Gs$ in ANN-PM and GPP that was used to estimate $Gs$ in conjunction with Medlyn's $Gs$ model in Medlyn-PM. We have the following conclusions.

1.  The optimal ANN architecture to estimate $Gs$ in ANN-PM consists of two hidden layers with 48 neurons in each layer, and that to estimate GPP in Medlyn-PM, two hidden layers and one neuron in each layer was optimal. The optimized $g_0$ and $g_1$ values in Medlyn's $Gs$ model are 0.06 and 3.94, respectively.
2.  The ANN-PM model can reasonably estimate the ET of cropland (RMSE = 19.09–19.50 W m$^{-2}$, MAE = 13.27–13.41 W m$^{-2}$, and R$^2$ = 0.83 for training, validation, and test datasets) and is proven to perform better than Medlyn-PM with a smaller RMSE and MAE and larger R$^2$.
3.  The ANN approach can represent the water stress impacts on ET well, as ANN-PM can reasonably capture the seasonal variations in ET at the dry sties (AI < 0.5). Additionally, the performances of the ANN-PM model at the dry sites were as good as at the wet sites.

**Author Contributions:** Conceptualization, Y.L. and Y.B.; Methodology, Y.L. and Y.B.; Software, Y.L. and Y.B.; Validation, Y.L.; Formal analysis, Y.B.; Investigation, Y.L. and Y.B.; Resources, Y.B.; Data Curation, Y.L. and Y.B.; Writing—Original Draft, Y.L.; Writing—Review and Editing, Y.B., S.Z., J.Z., and L.T.; Visualization, Y.L. and Y.B.; Supervision, Y.B.; Project administration, Y.B.; Funding acquisition, Y.B. and J.Z. All authors have read and agreed to the published version of the manuscript.

**Funding:** This research was funded by the National Natural Science Foundation of China (Grant Nos. 41,901,342, 31,671,585), "Taishan Scholar" Project of Shandong Province, and Key Basic Research Project of Shandong Natural Science Foundation of China (Grant No. ZR2017ZB0422).

**Institutional Review Board Statement:** Not applicable.

**Informed Consent Statement:** Not applicable.

**Data Availability Statement:** The data used in the study can be downloaded through the corresponding link provided in Section 2.4.

**Acknowledgments:** This work used eddy covariance data acquired and shared by the FLUXNET community, AmeriFlux, AsiaFlux, and European Flux Database Cluster. The FLUXNET also includes these networks: AmeriFlux, AfriFlux, AsiaFlux, CarboAfrica, CarboEuropeIP, CarboItaly, CarboMont, ChinaFlux, Fluxnet-Canada, GreenGrass, ICOS, KoFlux, LBA, NECC, OzFlux-TERN, TCOS-Siberia, and USCCC. The FLUXNET eddy covariance data processing, and harmonization was carried out by the ICOS Ecosystem Thematic Center, AmeriFlux Management Project and Fluxdata project of FLUXNET, with the support of CDIAC, and the OzFlux, ChinaFlux, and AsiaFlux offices.

**Conflicts of Interest:** The authors declare no conflict of interest.

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
