# Peer review of "Using Artificial Neural Network Algorithm and Remote Sensing Vegetation Index Improves the Accuracy of the Penman-Monteith Equation to Estimate Cropland Evapotranspiration"

_applsci, doi:10.3390/app11188649_

Round 1
Reviewer 1 Report
Summary
In this study, the authors compare two possible ways to combine machine learning methods with the Penman-Monteith (PM) to improve estimation of evapotranspiration in croplands. The study is interesting and appropriate for the journal but I have some concerns about the design of the study, the use of only 17 stations for training an ANN model, and the manuscript itself. The major concerns and some minor comments are detailed below.
Major Comments
- The authors compare two ‘hybrid’ models which are both based on the PM equation. However, they do not compare the models with the purely process-based PM model itself. I think this is an important requirement as any improvement seen in the addition of the ANN should be compared against the base model which is being improved.
- Infact, I do not understand the need for the Medlyn model. Why choose this one instead of the base PM model to compare against. The authors should provide a strong justification for this.
- Even if the Medlyn model can be justified, why use an ANN model to estimate GPP? I do not understand this. I am sure enough GPP datasets are available. It just adds one more layer of uncertainty. Again, I request the authors to provide strong justification for this.
- For training the ANN model, the authors consider various covariates (or input variables) but do not explain the reason for choosing any of them. For example, why was VPD or SW chosen?
- The number of stations is restricted to 17 stations. Are there enough data points to train a machine learning model? Even if it is well trained, is it enough to create a generalized model for Gs. In other words, can this model be used for other stations or regions? The authors need to discuss this.
- The authors make another unexplained choice: to conduct the study in dry climates. Why was this done?
Minor Comments
- The objectives stated in the Introduction are not necessarily research objectives (except point 4). The objectives should bring out why this study is being done and what issue is being addressed.
- The figures are of low quality. I request the authors to improve them.
- It would be good to show a map of the FLUXNET stations used in the study, to get an idea of the locations of these stations.
Reviewer 2 Report
SUMMARY
The paper addresses the research area related to the estimation of the cropland evapotranspiration by using the Penman-Monteith equation based on remote sensing data
It aims to improve the estimation of cropland ET by developing a hybrid ET model based on an artificial neural network and PM equation integrating meteorological data and earth observation data.
The author claim that this study confirmed the suitability of integrating ANN and PM equation to estimate ET of croplands under different environmental conditions and scale.
BROAD COMMENTs
As a general comment, the manuscript is fluent and well structured.
The author faces a new and interesting topic in the field of the estimation of ET by using hybrid models (ANN and PM eq.).
They claim, through a literature review, that the physical models or purely relied ML algorithms are not accurate enough to represent the ET due to limited ability to understand the ET process.
That’s the point. Nowadays, the use of ML models (hybrid or not) instead of physical models is the subject of debate in the scientific community, then in my opinion, in the paper, a comparison with a purely physical model should be used in order to conduct a robust validation.
MINOR COMMENTs
Figure6. Please consider improving the quality of the image. In particular, the labels on the x-axis are not easy to read.
Figure7. Please consider improving the quality of the image. The black lines (Observed ET) are not enough visible.
Reviewer 3 Report
The work is promising with good results in predicting ET in cropland but needs methodological improvements to facilitate understanding and clarity of the results.
Below are detailed the main points to be improved.
Abstract
The abstract should be a total of about 200 words maximum (see Instructions for Authors)
Introduction
In line 36 the citation [1] is from a 2017 publication, however the concept of evapotranspiration is from 1940. Moreover, this definition has already become popular. I suggest removing the citation replace it with other older quotations.
In the line 96 and 97 the sentence is misunderstood, because it says that the model is limited on a global scale, if it is global scale, it is not limitation. I believe what the authors mean is that it cannot be applied on a global scale? So the model is limited on a regional scale. Rewrite for clarity
In line 121 the authors mention three points, but there are four.
I suggest simplifying the objectives of the paper, there is no need for such specificity such as "Compare the accuracies of two models used to estimate the ET". Line 122-132
Methodos and Data
In line 133 I understand the authors' intention to change the name of the section, but the instructions for authors recommend using Material and methods. In addition it is used in thepaper the sensor MODIs and EC stations that I considered as material and not just a method or data.
Authors should insert the location where this study was conducted, as well as. For what locality were the MODIS images acquired? At what spatial resolution? Which MODIS pixel was selected? all pixels? Or only the one corresponding to the coordinate where the metrological stations are installed?
In line 148 it is mentioned that the accuracy of two models was evaluated just as it was mentioned in line 134-135. I suggest simplification to avoid redundancy, as it is showing lack of creativity to writing (I believe this is not the case of the authors)
In equation 1 it was missing to describe what ?E means
The term ML model development is used throughout the text, however the models are already developed, what is being done is ML model training. Replace the term ML model development with model training.
There is no need to explain the acronyms in Figure 2, as they have already been identified in Figure 1.
In section 2.1.1 ANN-PM model it is not demonstrated where the NDVI and NIRv data comes from. No mention is made of which MODIS bands, no study area and no spatial resolution
The authors used 7 predictor variables to estimate Gs, but no procedure was performed to check autocorrelation and no recursive filter elimination (RFE) were applied. Removing autocorrelated values will make the model cleaner, with fewer variables without compromising performance. RFE selects the least important variables for the model also by reducing the number of predictor variables without compromising the quality of prediction. This process makes the model easier to apply because it will require fewer predictor variables. (https://topepo.github.io/caret/recursive-feature-elimination.html, https://machinelearningmastery.com/rfe-feature-selection-in-python/, https://www.vertica.com/blog/in-database-machine-learning-2-calculate-a-correlation-matrix-a-data-exploration-post/)
The topic 2.4 Data should be inserted before training and evaluating the model's performance in order to follow a logical chronology. That is, before training, data selection is performed.
Results
In line 282 when calling Figure insert immediately after
In line 307 Table 5 is mentioned. The table or figure should be inserted immediately after it is called up in the text. In Table 5 is presented 4 approaches increasing the number of predictor variables from approach 1 to 4. These approaches should be inserted in the methodology so that the reader has prior knowledge of all predictor variables
Idem for Figure 4
It also shows the RMSE in percent, so you can get an idea of the error. Only the values of 18.82 or 23.72 I cannot understand if this is too much or too little!
Figure 4 is in very poor resolution. Improve!!!
In the text the ET intermediate time is kept, but in Figure 4 and 5 it is LE. It is understandable that LE represents ET, but I suggest standardizing it. Either use ET or use LE.
In the methodology the Akaike Information Criterion (AIC) is presented to measure the quality of the model, but in the results, I did not find the values found by AIC
Round 2
Reviewer 1 Report
The authors have satisfactorily addressed all my comments from the previous round of review.
Author Response
Thank you very much for the reviewers’ all comments on our manuscript from the previous round of review.
Reviewer 2 Report
Broad comment
The paper has been improved since the first round.
Minor comments
Figure 5. Please, consider specifying that "ET observed" has been measured from the flux tower.
Figure 7. Please, consider improving figure 7. (x-axis labels are too close to each other).
Figure 9. Idem
Reviewer 3 Report
Dear authors:
thank you for making the corrections.
All the figures in the results are in poor quality, which I believe may be caused by the journal's platform.
I suggest that you check it so that it is not published with such low quality.
